# Theoretical and Experimental Investigations on High-Precision Micro-Low-Gravity Simulation Technology for Lunar Mobile Vehicle

**DOI:** 10.3390/s23073458

**Published:** 2023-03-25

**Authors:** Weijie Hou, Yongbo Hao, Chang Wang, Lei Chen, Guangping Li, Baoshan Zhao, Hao Wang, Qingqing Wei, Shuo Xu, Kai Feng, Libin Zang

**Affiliations:** 1College of Mechanical and Vehicle Engineering, Hunan University, Changsha 410082, China; 2Tianjin Key Laboratory of Microgravity and Hypogravity Environment Simulation Technology, Tianjin Institute of Aerospace Mechanical and Electrical Equipment, Tianjin 300301, China; 3Beijing Institute of Control Engineering, Beijing 100190, China; 4Beijing Key Laboratory of Intelligent Space Robotic Systems Technology and Applications, Beijing Institute of Spacecraft System Engineering, Beijing 100094, China; 5School of Mechanical Engineering, Tianjin University, Tianjin 300350, China

**Keywords:** magnetic quasi-zero stiffness, lander, lunar vehicle, dynamics, micro-low-gravity simulation

## Abstract

With the development of space technology, the functions of lunar vehicles are constantly enriched, and the structure is constantly complicated, which puts forward more stringent requirements for its ground micro-low-gravity simulation test technology. This paper puts forward a high-precision and high-dynamic landing buffer test method based on the principle of magnetic quasi-zero stiffness. Firstly, the micro-low-gravity simulation system for the lunar vehicle was designed. The dynamic model of the system and a position control method based on fuzzy PID parameter tuning were established. Then, the dynamic characteristics of the system were analyzed through joint simulation. At last, a prototype of the lunar vehicle’s vertical constant force support system was built, and a micro-low-gravity landing buffer test was carried out. The results show that the simulation results were in good agreement with the test results. The sensitivity of the system was better than 0.1%, and the constant force deviation was 0.1% under landing impact conditions. The new method and idea are put forward to improve the micro-low-gravity simulation technology of lunar vehicles.

## 1. Introduction

Future manned and unmanned lunar base exploration missions put forward clear requirements for a multi-functional intelligent flying mobile vehicle carrier platform with assembly, exploration, and low-altitude flight capabilities [1]. Therefore, the new lunar vehicle (Figure 1) needs to break through key technologies, such as all-terrain active adaptation to the mobile mechanism, the planning and control of compound mobile mechanisms, and GNC (Guidance Navigation Control) flying at ultra-low altitudes to realize high-precision soft landing in complex terrain, to adapt to the harsh environment on the polar surface, and to take off repeatedly from the lunar surface [2].

The gravity field environment between the moon and the earth is very different [3]. Thus, the lunar vehicle carrier platform has more special characteristics compared with ground vehicles. It must pass mechanical properties and mobility performance tests such as landing buffer, passing performance, stability performance, obstacle-crossing performance, turning performance, and driving smoothness [4]. In the research field of micro-low-gravity simulation tests of planetary vehicles, researchers from the United States, Russian Federation, and China have made many attempts and explorations. The low-gravity experimental system of NASA (National Aeronautics and Space Administration) manned planetary vehicles is an active system [5,6]. A mechanism similar to the crown block is used to realize the following movement in a large range, and the two-dimensional servo platform is used to realize the accurate following movement in a small range. The active tracking constant tension suspension scheme was also adopted in the ground experiment of a Russian planetary vehicle [7]. The force balance method was mainly used in the ground test of China Chang’E Series detectors, which puts forward the unloading form based on a single sling [8]. In the buffer landing test of the Chang’E-III, the multi-rigid-body low-gravity simulation and simulation accuracy made new progress [9]. In the infield test of the lunar vehicle, methods such as adding a counterweight on the rocker’s arm and adjusting the suspension spring stress point were adopted [10]. As a result, the force between each wheel of the patrol car and the ground is the same as that on the lunar surface in various working conditions such as plane driving, climbing, and obstacle crossing. However, the shaking interference of the suspension rope system, especially when the vertical direction is under high dynamic conditions, has a great influence on the dynamic characteristics of the vehicle due to the large unloading error of the constant-force suspension. It has a significant impact on test accuracy under high-speed and high dynamic conditions. Therefore, under high dynamic conditions, high-precision constant force unloading is still a difficult problem in the micro-low-gravity simulation field at home and abroad [11,12]. The quasi-zero stiffness system is a kind of mechanism with high static stiffness and low dynamic stiffness characteristics. The characteristics of quasi-zero stiffness with high carrying capacity and low dynamic stiffness can be obtained through a specially designed mechanism in a certain stroke [13], which can solve the problem that the high stiffness of ordinary elastic element affects the accuracy of tension output. Carrella [14] proposed a method to obtain the quasi-zero stiffness mechanism in parallel with three springs. Tang [15] used the method of parallel positive and negative stiffness to obtain the improved quasi-zero stiffness mechanism in a three-spring parallel. Niu [16] proposed a tandem-tensioning constant torque mechanism to replace the buffer spring, which improved the mechanism significantly, increased the system bandwidth, and achieved an enhanced dynamic constant-force performance. Thus, the application of quasi-zero stiffness system in the field of gravity compensation provides a good solution. Whereas, in the mechanical contact quasi-zero stiffness system, there is inevitably friction between contact pairs [17]. For these reasons, the magnetic quasi-zero stiffness has the characteristics of no friction, large load, and fast dynamic response [18,19,20], which provides a new idea for the constant supporting force with a heavy load, long stroke, and high precision in the vertical direction.

In view of the test requirements of the future flying mobile vehicle platform, how to further improve the simulation accuracy of vehicle dynamics and improve the unloading accuracy when the vehicle moves rapidly has become a key issue. Therefore, this paper focuses on the high dynamic constant force unloading technology in the vertical direction. In this paper, a new three-dimensional high-precision and micro-low gravity simulation test system based on magnetic quasi-zero stiffness was designed. A position control method based on fuzzy PID (Proportion Integration Differentiation) parameter tuning was proposed, and a gravity balance system with high precision, large range, and fast response was realized. The performance of the prototype was tested and analyzed by simulation and experiment, which can provide a useful attempt for the test of intelligent flying mobile vehicle carrier platforms in a manned/unmanned lunar base in the future.

The structure of this paper is as follows: the theoretical model of the micro-low-gravity simulation system for a lunar vehicle and the research methods are shown in Section 2. Section 3 formulates the prototype of the vertical constant force mechanism and the evaluation experiment. Section 4 focuses on the analysis and discussion of the landing buffer simulation experimental results. The conclusion is given in Section 5.

## 2. Theoretical Model and Methods

### 2.1. Design of Micro-Low-Gravity Simulation System for Lunar Vehicle

The schematic architecture of the lunar vehicle micro-low-gravity simulation test system is shown in Figure 2a,b. The micro-low-gravity simulation system can realize the high-precision simulation of the lunar vehicle landing process (Figure 2c) and the self-adaptive height adjustment during the horizontal movement with the lunar vehicle (Figure 2d). During the micro-low-gravity simulation test, a three-free air floating ball bearing is used to adjust the attitude of the lunar vehicle to achieve horizontal balance. In order to ensure that the force between the wheel and the ground is the same as the lunar surface in various working conditions, such as plane driving, climbing, and obstacle crossing during the landing buffer and patrol, a magnetic quasi-zero stiffness mechanism and a stroke amplification mechanism are introduced to realize vertical unloading.

### 2.2. Principle of Magnetic Quasi-Zero Stiffness

The magnetic quasi-zero stiffness (M-QZS) mechanism is based on the “repulsion” magnetic negative stiffness principle of permanent magnets and is formed by connecting the main bearing positive stiffness subunit and the magnetic negative stiffness subunit in parallel [21]. Figure 3 shows the principle of magnetic quasi-zero stiffness unit. As the main bearing unit, the positive stiffness mechanical spring bears the static load. The relation of the quasi-zero stiffness system is shown as Formula (1).
(1)$$\left\{ \begin{array}{l} k_{1}>0\\ k_{2}<0\\ k=k_{1}+k_{2}\approx 0 \end{array}\right.$$
where *k*_1_ is the stiffness of the mechanical spring, *k*_2_ is the stiffness of the magnetic spring, and *k* is the stiffness of the system.

When the mechanism is in a balanced state, as is shown in Figure 3a, the supporting force provided by the positive stiffness spring is equal to the gravity generated by the static load, which satisfies the following formula.
(2)$$F_{1}=mg$$
where $F_{1}$ is the main load-bearing positive stiffness subunit force, as is shown in Figure 3, *m* is the mass of the load, and g is the acceleration of gravity on the Earth.

When the downward external force *F*_i_ is applied to the static load, as is shown in Figure 3b, the main load-bearing positive stiffness subunit force *F*_1_ increases upward, and the output force *F*_2_ generated by the negative stiffness magnetic spring increases downward. For this reason, the mechanical stiffness and magnetic stiffness are satisfying Equation (2) within a certain range of motion. Then,
(3)$$F_{1}^{\prime }=mg+F_{2}$$

At the same time, the friction resistance can be neglected because of the air-floating guidance. Therefore, non-contact magnetic zero stiffness overcomes the mechanical friction defect of the mechanical negative stiffness spring and has the advantages of compact structure and adjustable stiffness.

According to the model of magnetic charge [20,22], the stiffness of the magnetic spring can be obtained by Equation (4).
(4)$$K_{S}=\frac{JJ^{\prime }}{2\pi \mu_{0}}{\sum_{i=0}^{1}\sum_{j=0}^{1}\sum_{k=0}^{1}\sum_{l=0}^{1}\sum_{p=0}^{1}\sum_{q=0}^{1}\left(-1\right)}^{i+j+k+l+p+q}\varphi (L_{ij},M_{kl},N_{pq},r)$$
with
(5)$$\varphi (L_{\mathrm{ij}},M_{\mathrm{kl}},N_{\mathrm{pq}},r)=r+M_{\mathrm{kl}}\ln(r-M_{\mathrm{kl}})$$

The variables can be calculated according to Equation (6).
(6)$$\{\begin{array}{l} L_{ij}=\Delta x+{\left(-1\right)}^{j}A/2-{\left(-1\right)}^{i}a/2\\ M_{kl}={\left(-1\right)}^{l}B/2-{\left(-1\right)}^{k}b/2\\ N_{pq}=d+{\left(-1\right)}^{q}C/2-{\left(-1\right)}^{p}c/2\\ r=\sqrt{L_{ij}^{2}+M_{kl}^{2}+N_{pq}^{2}} \end{array}$$
where J and J′ are the magnetic polarization intensity vector, and the magnetic springs design parameters *A*, *B*, *C*, *a*, *b*, *c*, and *d* were shown in Figure 3. Δx is the deformation of the supporting unit, which can be calculated by $\Delta x=x_{ve}-x_{sy}$ as shown in Figure 4.

### 2.3. M-QZS Vertical Constant Force System

#### 2.3.1. Dynamic Model

Figure 4 shows the schematic diagram of the M-QZS vertical constant force support system for the lunar vehicle gravity compensation. The magnetic quasi-zero stiffness mechanism is connected in series with the stroke amplification mechanism, in which the M-QZS can output constant force in a small stroke range, and the floating support is actively controlled by the servo system to amplify the constant force stroke.

When the lunar vehicle is in the lunar and earth field environment, the vertical dynamic can be formulated as Equations (7) and (8), respectively.
(7)$$m_{ve}\ddot{x}=F_{e}-m_{ve}g_{lu}$$
(8)$$m_{ve}\ddot{x}=F_{e}-m_{ve}g$$
where, $m_{ve}$ is the mass of the lunar vehicle, $F_{e}$ is the external force on the spacecraft, $g_{lu}$ is the lunar gravity, and g is the Earth’s gravity; therefore, $g_{lu}=g/6$.

Therefore, when the lunar vehicle is simulated on the ground in the gravity field of the earth, it is necessary to provide a compensation force to offset part of the gravity generated by the Earth. Regardless of the transmission mass and the damping of the air, the dynamic model of the lunar vehicle is formulated as follows
(9)$$m_{ve}\ddot{x}=F_{e}-(m_{ve}g-F_{com})$$
where, $F_{com}$ is the compensation force.

The dynamic characteristics of the vertical constant force support system in the vertical direction meet the following conditions.
(10)$$\{\begin{array}{l} m_{ve}{\ddot{x}}_{ve}=F_{e}-m_{ve}g+F_{0}+F_{\Delta }\\ m_{sy}{\ddot{x}}_{sy}=F_{sy}+F_{dsy}-m_{sy}g-F_{0}-F_{\Delta } \end{array}$$
where $F_{0}$ is the ideal value of the system supporting force.

The dynamic model of simulated load is shown in Formula (11) under the real zero-gravity environment.
(11)$$F_{e}=m_{ve}\ddot{x}+c({\dot{x}}_{ve}-{\dot{x}}_{sy})+k(x_{ve}-x_{sy})$$
where, $x_{ve}$ is the vertical displacement of the lunar vehicle, and $x_{sy}$ is the displacement of the servo system.

Since there is no influence of contact friction, the change of supporting force of the constant force system can be determined by Formula (12).
(12)$$F_{\Delta }=k(x_{ve}-x_{sy})+c({\dot{x}}_{ve}-{\dot{x}}_{sy})$$

In the analysis, the constitutive model of the M-QZS is simplified as a parallel combination of a constant force and a nonlinear spring, and the constant force is equal to the gravity of the simulated load carried by the M-QZS. By measuring the stiffness–displacement data of the M-QZS under a static load, the stiffness–displacement curve is drawn in Figure 5. The stiffness K displacement equation of the M-QZS is fitted by the least square method as shown in Formula (13). It can be seen from Figure 5 that the stiffness of the zero stiffness mechanism is lower than 8 N/mm within the range of ±3 mm, and the stiffness of the system increases obviously with the increase in compression.
(13)$$r=0.6715x^{3}-2.405x^{2}+1.787x+6.712$$

#### 2.3.2. Control System Design and Active Control Strategy

In the closed-loop control constant force system, the control strategy of the force closed loop or the force potential coupling is usually adopted. That is, the force and position sensor is used to directly measure the force signal or simultaneously detect the force signal and displacement signal. However, the sensitivity of the force sensor is inversely proportional to its range, that is, as the range increases, the sensitivity decreases [23]. With the accuracy of 0.5% of the high dynamic force sensor, when the pressure reaches 5000 N, the measurement error of the force sensor will reach more than 25 N. Thus, the error is hard to ignore. On the other hand, due to the sudden change of input force signal in the process of lunar vehicle landing buffer, the rapidity of the active following control of the stroke amplification mechanism is extremely demanding. Therefore, the closed-loop control of the position information was adopted in this system instead of the traditional closed-loop control of the force information.

A position control method based on fuzzy PID parameter tuning was proposed for the system. Figure 6 shows the block diagram of the micro-low-gravity simulation test system control strategy. In this system, the M-QZS is connected in series with the stroke amplification mechanism. Due to the M-QZS keeping a low stiffness in a certain range while carrying heavy loads, the relative displacement difference between the simulated load and the floating support is directly measured with a high-precision grating ruler. The Δx is used as feedback to participate in the control closed loop of the stroke amplifying mechanism. *G_C_*(*s*) is the transfer function of the controller, and *G_D_*(*s*) is the transfer function of the stroke amplifying mechanism, as can be seen from Figure 6. When Δx approaches zero, that is, $x_{ve}$ is equal to $x_{sy}$, this can ensure that the constant force system can truly simulate the micro-low-gravity environment. At the same time, because the sudden change of force will not affect the displacement instantly, the change in high-frequency force is responded to by the M-QZS, which greatly reduces the speed requirement of the stroke amplification mechanism.

The constant force controller adopts a fuzzy self-adaptive tuning PID control algorithm, as shown in Figure 7. The controller takes the position error as input, and uses fuzzy to modify PID parameters online. A multi-input and multi-output fuzzy controller were adopted to fuzzify the position signal of the system and input it to the controller. We set the fuzzy sets of variables Δ*x*, Δ*F*, and the control quantities Δ*Kp*, Δ*Ki*, and Δ*Kd* in the operation of the simulation system as NB (negative big), NM (negative middle), NS (negative small), ZE (zero), PS (positive small), PM (middle) and PB (fair).

### 2.4. Dynamic Simulation of Micro-Low-Gravity Simulation System

The three-dimensional solid model of the M-QZS vertical constant force support system was established by Solidworks, and then the system model was imported into SimMechanics to obtain the inertia and mass parameters of each component. Then, define the motion constraint relationship between each component of the system. According to the measured displacement–stiffness data of the zero-stiffness mechanism, the spring damping model of the magnetic zero-stiffness mechanism was established in SimMechanics. The mechanical simulation model of the system was completed, and the input parameters of the simulation test are shown in Table 1.

## 3. Design and Performance Evaluation Experiment

### 3.1. System Architecture

#### 3.1.1. Structure Composition

As is shown in Figure 8, the prototype of the vertical constant force mechanism in the micro-low-gravity simulation system of lunar vehicles was mainly composed of four parts: the M-QZS constant force support system, the simulated landing surface, the stroke amplification mechanism, a grating ruler, and an integrated control system. The air bearing was used in the M-QZS, which makes the negative stiffness rotor move frictionless in the vertical direction. The stroke amplification mechanism was composed of a polished lead screw guide rail, and, at the same time, a guide rod was installed on the landing surface. The above design ensured the vertical movement during the simulated buffer test.

The stroke amplifying mechanism was mainly composed of a servo motor, a timing belt, a turn screw, and a linear guide rod. The motor SGD7S-550A made by Yaskawa Electric was selected as the severing motor, with detailed parameters shown in Table 2. The transmission ratio of the synchronous belt was 1:1. The maximum vertical movement speed of the stroke amplifying mechanism was 1250 mm/s, and the positioning accuracy and repeated positioning accuracy of the mechanism were better than 1 μm. The grating ruler (VIA-0100, MicroE Systems Co. Ltd., Natick, MA, USA), which has a resolution of 0.1 μm, was utilized to measure the deformation of constant force system. A spring with vertical guidance was installed on the landing surface, which was used to simulate the rebound phenomenon of the lunar vehicle in the landing process.

#### 3.1.2. Integrated Control System

The integrated control system transformed the windows system into a hard real-time system by installing the RTX (Real Time eXecutive) real-time system on the Windows 7 system (32-bit and 2016 version). Among them, a MFC man–machine interface ran on the Windows system, and the servo control program ran on the RTX system in real time. The clock resolution of the RTX can reach 100 nanoseconds, and the minimum timer period can be 1000, 500, 200, and 100 microseconds, thus ensuring the high-speed operation of the servo algorithm. There are four cores in the CPU. When installing the RTX system, two cores were assigned to the RTX system and two cores to windows system. The minimum real-time cycle unit of the RTX system was 10 microseconds. In this study, the running cycle of the RTX system was set to 1 ms, which meets the requirements of the control system. The human–machine interface is shown in Figure 9.

### 3.2. Test Method

The working conditions of the lunar vehicle landing buffer simulation test in the micro-low-gravity environment are shown in Table 3. The mass of the lunar vehicle was 94.5 kg. The constant force on the test object was provided by a weight to simulate the dynamics of the lunar vehicle in the micro-low-gravity environment. In order to realize the simulation of different acceleration conditions from zero gravity to microgravity, the external force of 0.98~196 N was selected, which can be equivalent to the theoretical gravity acceleration of 0.00106~0.17 g. When the external force was 196 N, the equivalent gravity acceleration was about 1/6 g, which is the working condition of lunar gravity acceleration. The mass of the weights was 0.1 kg, 1 kg, 10 kg, and 20 kg, which simulate the constant force of 0.98 N, 9.8 N, 98 N, and 196 N, respectively. The travel amplifying mechanism was raised to 170 mm from the landing surface, and the system was in a balanced state at this time. The acceleration value was calculated by Formula (14).
(14)$$a_{ve}=\frac{m_{e}g}{m_{ve}+m_{e}}$$
where $m_{e}$ is the mass of the weight.

During the experiment, the integrated control system ran with a servo cycle of 1 ms. The grating with a resolution of 0.001 mm was used to measure the relative displacement difference, which is the input of the control algorithm of the stroke amplification mechanism. The acceleration of the test object landing process was collected in real time through the high-precision inertial navigation unit, the model of which was IMU330A, which had an acceleration resolution of 0.07 mg. The displacement of the stroke amplification mechanism during the whole falling process was measured through the rotary encoder provided by the servo motor, which had a displacement measurement resolution of 0.006 mm.

## 4. Results and Discussion

### 4.1. Simulation Results

Figure 10 shows the simulation results based om the working conditions in Table 1. It can be seen from Figure 10a,b that, during the falling process, the simulated load was uniformly accelerated, and the displacement curve was parabolic. As can be seen from Figure 10c, the deformation displacement Δx was 0.06 mm in the falling process. According to Equation (12), the constant force deviation was 0.04 N, as is shown in Figure 10d. The constant force accuracy was 0.04% in the uniform acceleration stage and 0.08% in the collision landing stage, which shows that the control strategy could achieve high constant force accuracy in the simulation stage and meet the requirements of subsequent tests.

### 4.2. Experimental Results

Figure 11, Figure 12 and Figure 13 show the results of the landing buffer test with 0.98 N, 9.8 N, 98 N, and 196 N equivalent external force, respectively. It can be seen obviously from the displacement and velocity curve in Figure 11 that the simulation object moved smoothly in different external force. Within a few tens of seconds, the simulator repeatedly rebounded to the ground. In the fall time of 20~60 s, the simulator bounced off the ground several times under different external forces. From the test results of the acceleration curve in Figure 13d, it can be seen that the acceleration of the simulator was about 0.17 g in a steady state when it fell, which is the exact equivalent of the gravity acceleration in lunar conditions. The test system reproduced the phenomenon that the lunar vehicle bounces when it falls in the micro-low-gravity simulation environment.

As is shown in Figure 12a–d, the max velocity of the simulation object was 57.37 mm/s, 152.53 mm/s, 551.67 mm/s, and 647.81 mm/s with 0.98 N, 9.8 N, 98 N, and 196 N equivalent external force, respectively. With the increase in the weight, the speed and acceleration of the test object increased. Figure 13a–d show the acceleration results of the test object with different equivalent external forces. It can be seen that the acceleration curve fluctuated in a small range around the theoretical acceleration value of the experimental object, which is a blue dotted line in the figure, during the falling process.

### 4.3. Discussion

The key parameter pairs of simulation and test are shown in Table 4. As can be seen from the simulation and test results, the coincidence degree between the falling acceleration and the maximum speed parameter of the simulation and the test was more than 90%. The coincidence degree of the first rebound period of the simulation and the test was 98.2%, and the coincidence degree of the first rebound peak was 99%, which well represents the consistency between the simulation and the test. It can be shown that the simulation system could accurately verify the accuracy of the control model.

In order to further analyze the constant force accuracy of the vertical constant force support system under different equivalent external forces, the constant force deviation curve under four working conditions was compared, as is shown in Figure 14. For the test object of 956 N, the system could accurately identify the external force of 0.98 N, and the constant force deviation was only 0.04 N. With the equivalent lunar gravity acceleration, that is, the external force was 196 N, the constant force deviation was 1.9 N. The constant force error was less than 0.1%. These results proved that the gravity compensation method based on magnetic quasi-zero stiffness and the designed prototype was very sensitive to small forces. Compared with the unloading accuracy of the traditional mechanical constant system and cylinder constant force system, the unloading accuracy of the M-QZS constant force system reached 0.1%, which is higher than the mechanical constant force unloading accuracy of 0.5% [24,25] and the cylinder constant force unloading accuracy of 0.2% [26].

What’s more, it can be also found that, with the increase in external force, the deviation of constant force increased slightly. The main reason is that the deformation of the M-QZS system and the dynamic response speed increased with the increase in external force. Therefore, it is necessary to further improve the response accuracy of the servo following system under heavy load. The experiment of this study was carried out in normal temperature and atmospheric environment, and the influence of wind resistance was ignored. For future spacecraft with large volumes and high dynamic motion, it is also necessary to consider the unloading behavior under vacuum conditions when the air resistance is difficult to ignore.

## 5. Conclusions

In this paper, a high-dynamic and long-range constant force supporting system based on magnetic quasi-zero stiffness for lunar vehicles was designed. The results include the following:
The non-contact/frictionless M-QZS mechanism was combined with the stroke amplification structure, which realized high-precision and high-dynamic constant force maintenance in the vertical direction. It provided a new method for the micro-low-gravity simulation test of lunar vehicles on the ground.A position control method based on fuzzy PID parameter tuning was proposed, and the dynamic characteristics of the system were simulated and analyzed using joint simulation technology, which verified the feasibility and accuracy of the control model.The prototype of the micro-low-gravity simulation test system for the lunar vehicle was built, and landing buffer tests under different external forces were carried out. The results show that the system had good stability and high accuracy under different external loads. The force sensitivity was better than 0.1%, and the constant force error was less than 0.1%. The test object bounced up many times during the landing buffer test, which successfully reproduced the phenomenon of the lunar vehicle touchdown rebound during landing in a slight micro-low-gravity environment.Later, the research on vertical adaptive adjustment during horizontal movement of the lunar vehicle will be carried out. The performance of the test system to maintain vertical constant force accuracy will be further verified when the lunar vehicle walks on the undulating road surface in the next step.

## Figures and Tables

**Figure 1 sensors-23-03458-f001:**
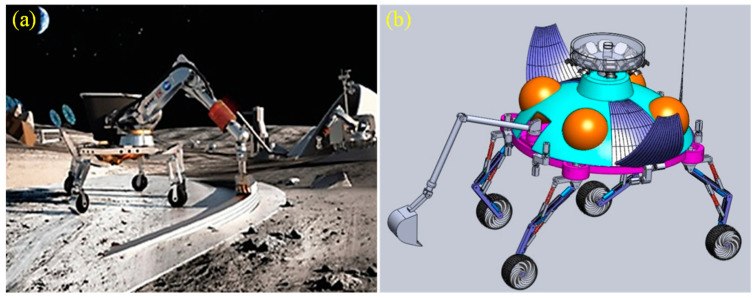
The lunar vehicle. (**a**) physical diagram, (**b**) three-dimensional model diagram.

**Figure 2 sensors-23-03458-f002:**
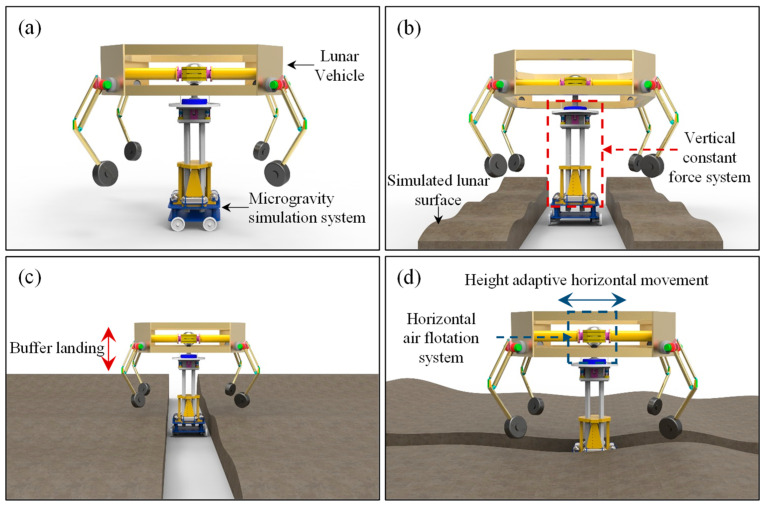
Three-dimensional schematic diagram of micro-low-gravity simulation system structure. (**a**) system architecture, (**b**) lunar vehicle infield test, (**c**) buffer landing simulation, (**d**) horizontal movement simulation.

**Figure 3 sensors-23-03458-f003:**
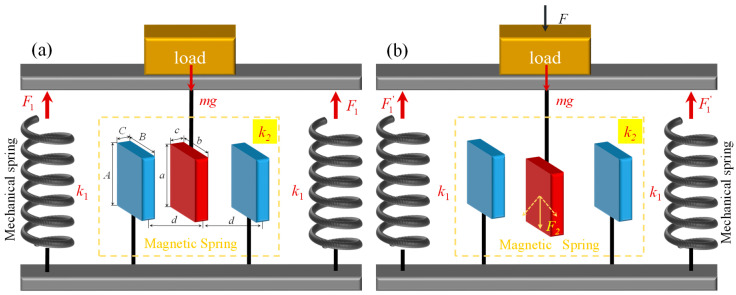
Principle of magnetic quasi-zero stiffness. (**a**) ateady state, (**b**) applying external load.

**Figure 4 sensors-23-03458-f004:**
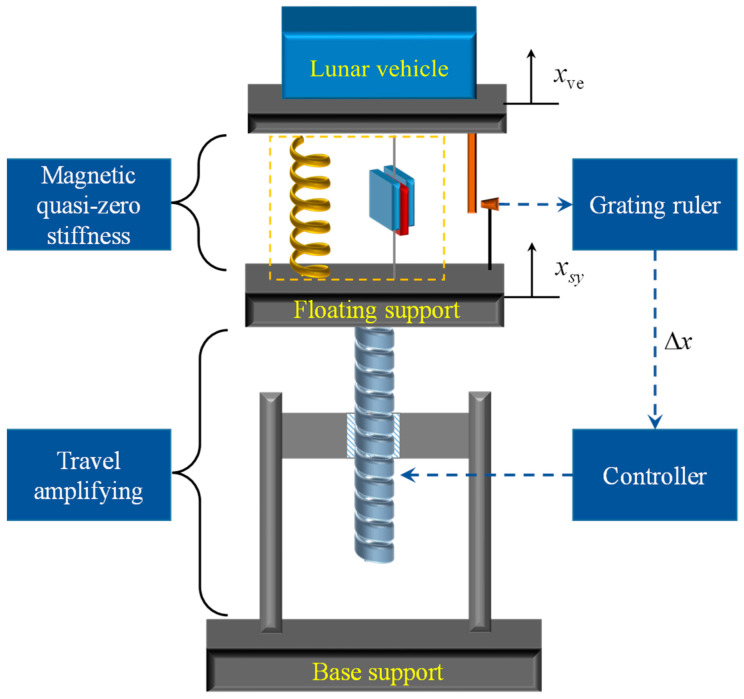
Schematic diagram of the lunar vehicle test system.

**Figure 5 sensors-23-03458-f005:**
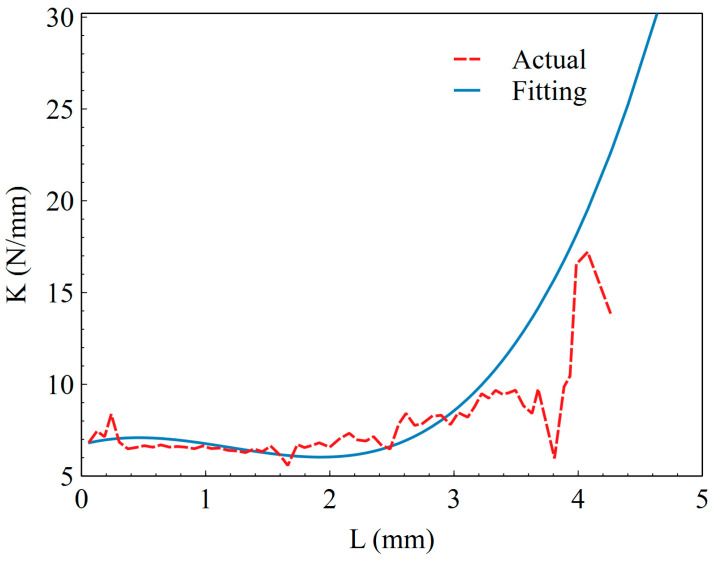
Stiffness–displacement curve of magnetic zero stiffness mechanism.

**Figure 6 sensors-23-03458-f006:**
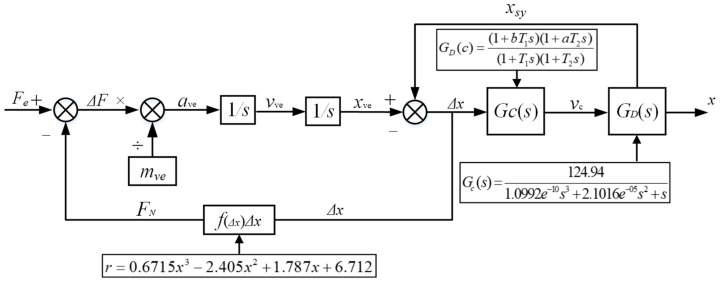
Active control strategy block diagram.

**Figure 7 sensors-23-03458-f007:**
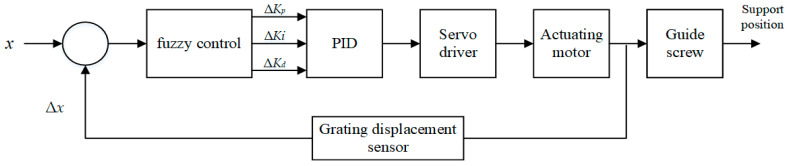
Constant force control principle based on fuzzy PID.

**Figure 8 sensors-23-03458-f008:**
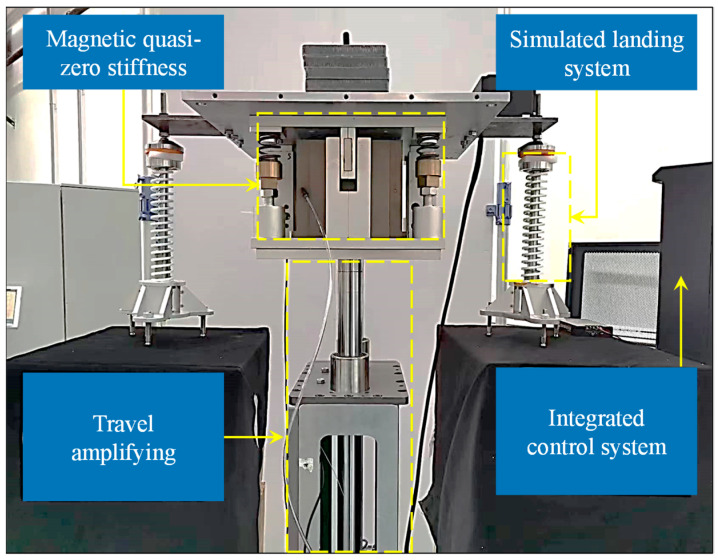
Prototype of landing buffer simulation test system for lunar vehicle.

**Figure 9 sensors-23-03458-f009:**
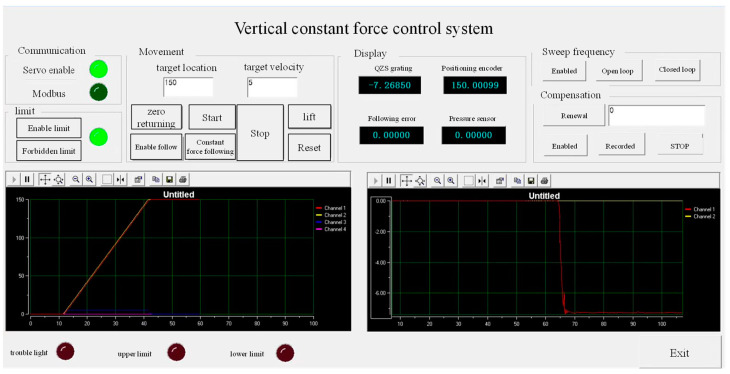
Man–machine interface of M-QZS vertical constant force.

**Figure 10 sensors-23-03458-f010:**
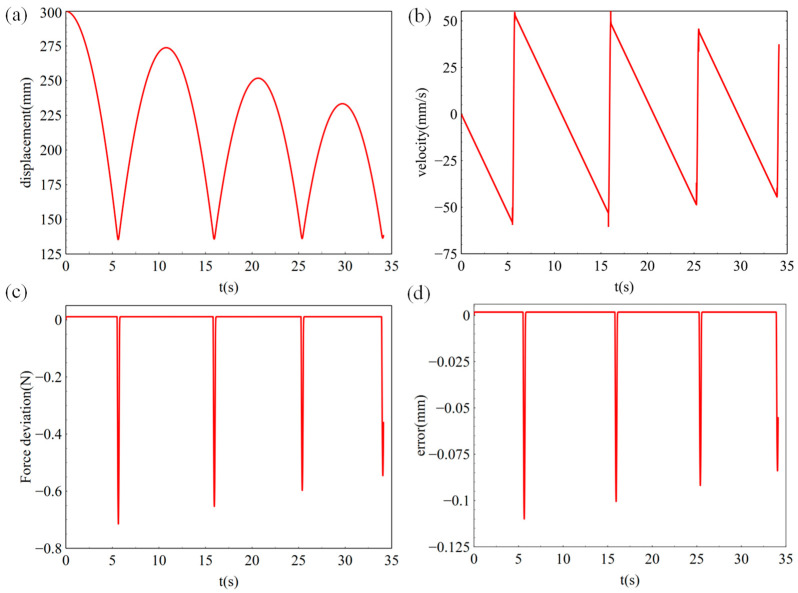
Simulation results of the landing buffer tests with input force of 0.98 N. (**a**) re-displacement curve, (**b**) velocity curve, (**c**) force deviation, (**d**) following error.

**Figure 11 sensors-23-03458-f011:**
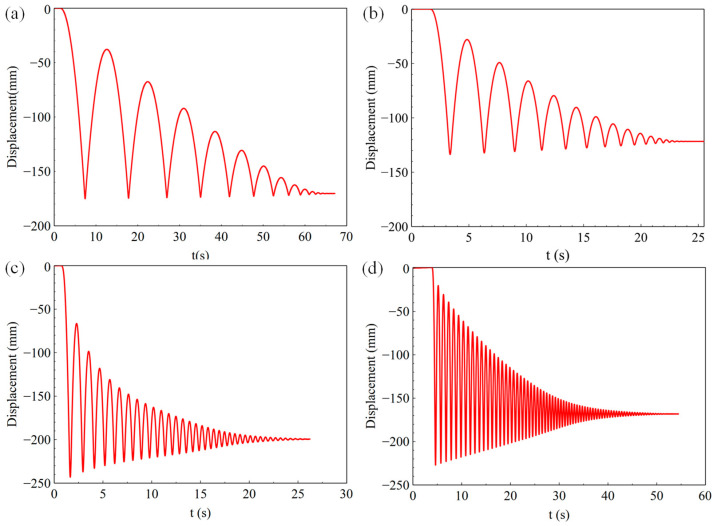
Displacement of the landing buffer tests with external force of (**a**) 0.98 N, (**b**) 9.8 N, (**c**) 98 N, (**d**) 196 N.

**Figure 12 sensors-23-03458-f012:**
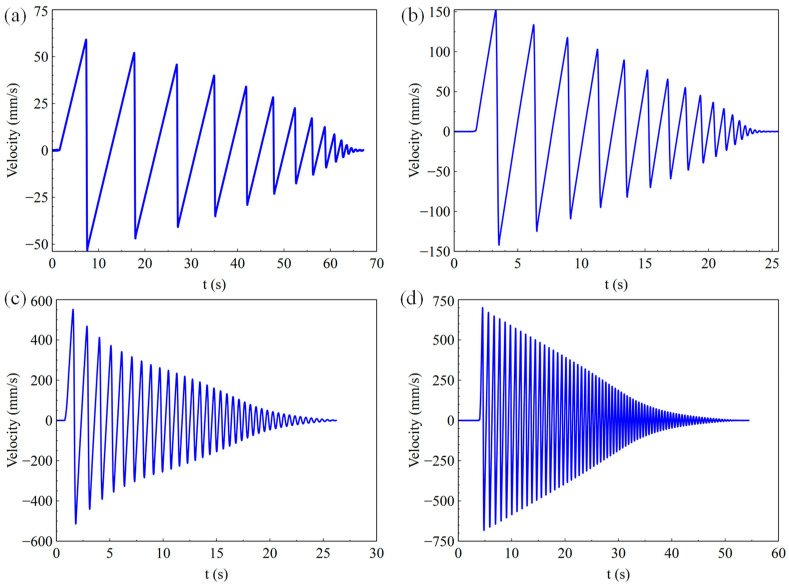
Velocity of the landing buffer tests with external force of (**a**) 0.98 N, (**b**) 9.8 N, (**c**) 96 N, (**d**) 196 N.

**Figure 13 sensors-23-03458-f013:**
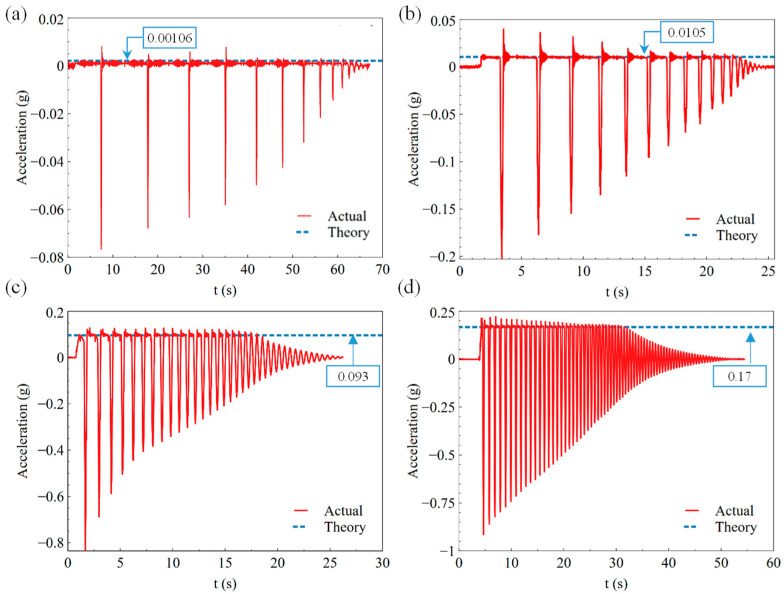
Acceleration of the landing buffer tests with external force of (**a**) 0.98 N, (**b**) 9.8 N, (**c**) 96 N, (**d**) 196 N.

**Figure 14 sensors-23-03458-f014:**
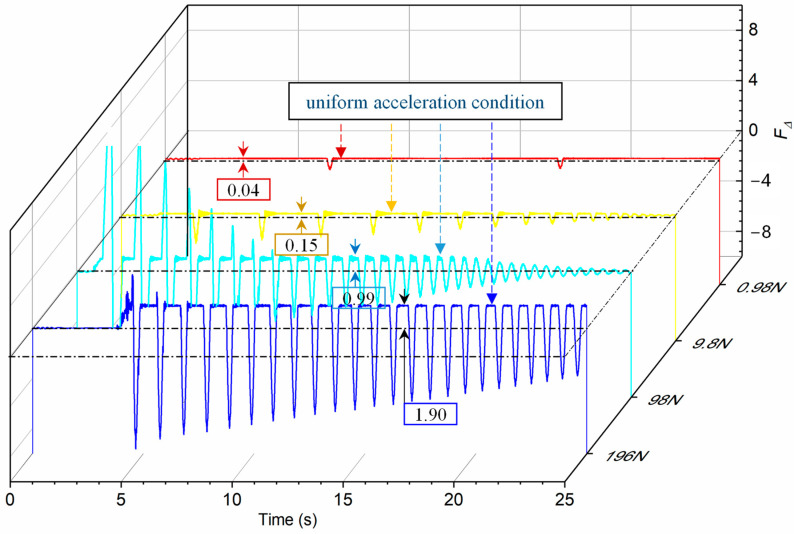
Constant force deviation of the landing buffer tests with external force of 0.98 N, 9.8 N, 96 N, 196 N.

**Table 1 sensors-23-03458-t001:** The input parameters for simulation test.

Parameters	Value
external force (N)	0.98
mass of lunar vehicle (kg)	94.5
falling height (mm)	170

**Table 2 sensors-23-03458-t002:** The parameters of the stroke amplifying mechanism.

Parts	Parameters	Value
Servo motor	Power (kW)	7.5
	Torque (Nm)	48
	rated speed (rpm)	1500
Turn-screw	Stroke (mm)	480
	Lead (mm)	50

**Table 3 sensors-23-03458-t003:** The input parameters of landing buffer test.

Parameters	Value
mass of the weight (kg)	0.1
	1
	10
	20
mass of the lunar vehicle (kg)	94.5
falling height (mm)	170

**Table 4 sensors-23-03458-t004:** Comparison of simulation and test results with external force of 0.98 N.

Falling Parameters	Simulation	Experimental
Acceleration (g)	0.001058	0.00106 ± 0.001
Max velocity (mm/s)	58.504	59.37
First rebound period (s)	10.59	10.39
First rebound amplitude (mm)	264.73	262.15

## Data Availability

Not applicable.

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
