# Peer review of "Theoretical and Experimental Investigations on High-Precision Micro-Low-Gravity Simulation Technology for Lunar Mobile Vehicle"

_sensors, 2023, doi:10.3390/s23073458_

Round 1
Reviewer 1 Report
This paper presents a high-precision and high-dynamic landing buffer test method based on the principle of magnetic quasi-zero stiffness. The introduction states the main purpose of the paper, and the relation between the paper and the previous works is clearly explained. The efficiency of the presented method is illustrated via experiments with a prototype of the lunar vehicle constant force support system.
The following are my specific comments:
(1) For clarity, the authors are suggested to provide a specification of “quasi-zero stiffness” in the introduction.
(2) How can we keep the balance of the lunar vehicle micro-low gravity simulation test system shown in Figure 2? It is better to provide some explanations.
(3) I cannot find the definition of the denotation “M” in equation (5) and “F0” in equation (10). In addition, the full names of the abbreviations should be given when they appear firstly in the paper, such as "M_QZS".
(4) The authors are suggested to illustrate the high performance of the presented landing buffer test method in comparison with other existing methods via simulations.
(5) Why is the constant force selected as 0.98N, 9.8N, 98N and 196N? Is it related to the working conditions of the lunar vehicle? It is better to provide some explanations.
(6) Some descriptions about the reproduction of the phenomenon of the lunar vehicle touchdown rebound in the experiment should be given.
(7) Will the horizontal movement of the lunar vehicle affect the performance of the position control method in the vertical direction? Try to provide some remarks.
Reviewer 2 Report
The article entitled "Theoretical and Experimental Investigations on High-Precision Micro-low Gravity Simulation Technology for Lunar Mobile Vehicle" introduces a high-precision microgravity simulation technology for lunar rover. The subject area of the research is very interesting and it does hold some significance in the field of research. However, there are some concerns that must be improved before this article can be accepted for publication in the journal Sensors from MDPI. The detailed concerns are listed below:
1) The literature review on high-precision constant force unloading is limited. I suggest using recent journal articles to expand the literature review.
2) The variables in the equations are not described throughout. For example, the meaning of F0 in formula (10). Ideally, the variables of the equations are described right after the introduction of the equation. This makes the explanation even difficult to follow.
3) The formula layout of lines 126, Equation (7), (8), (9), (10) needs to be unified
4) It is recommended to explain Figure 6 and the variables in more detail.
5) The format of Table 1 and Table 2 need to be adjusted, some contents are not centered.
6) The novelty to the paper is not very clear because there is no comparison between the results and previous studies. The author need to highlight this paper's innovative contributions.
7) The design of the experiment is not comprehensive enough. In this paper, the landing buffer tests under different external forces are carried out, and the stability and accuracy of the system are proved. However, there are not enough experiments to support the conclusion of "... successfully reproduced the phenomenon of the lunar vehicle touchdown rebound" in line 325. The system in this paper is designed for the lunar rover, but it does not introduce the effectiveness of the simulation of lunar microgravity.
Reviewer 3 Report
This paper introduces a high-precision simulation technology of micro-low gravity on the ground for lunar vehicle, which is very interesting and meaningful. It could be published subject to the revisions below:
(1) The landing buffer test for the lunar vehicle is conducted under four working conditions: external force of 0.98 N, 9.8 N, 96 N, 196 N. What is the basis for selection? What kind of working condition is described by the application of lunar gravity acceleration working condition?
(2) This paper introduces the vertical unloading system of lunar vehicle in detail, however, how to keep the vertical direction during the landing buffer process? It is suggested to supplement the setting of test state and explain the problem of vertical maintenance during landing.
(3) In fact, the lunar vehicle not only has vertical motion during working state, but also may have pitching movement. How to consider the micro-low gravity simulation test of other degrees of freedom?
Reviewer 4 Report
The topic of the manuscript is interesting given fits the scope of the Journal. After a careful revision, the following comments are provided for the enhancement of the manuscript.
- Regarding the format of the document, some suggestions are as follows.
The acronym GNC should be defined in line 36. The same occurs for PID, despite of being a commonly term in control systems, it should be decomposed the first time that it is used.
Also, RTX stands for Real Time eXecutive, which appears in line 224.
- The paper is well written and organized; about the content of the manuscript, these issues are commented.
The structure of the manuscript should be indicated at the end of the introductory section. This is a common practice of research papers that enhances readability.
In the subsection 2.3, more details about the fuzzy PID must be provided. For example, a flowchart of the conducted steps for parameter tuning could enrich this aspect.
In the subsection 3.1.2, concerning the RTX real-time system, the specific version should be given due to the fact that the version of Windows is a bit old. The amount of RAM memory and the CPU could also be indicated. In a similar sense, more details about the MFC man-machine interface should be provided for a whole description of the simulation approach. In fact, a block diagram depicting the simulation environment linkages would be illustrative.
Figure 12 is very illustrative.
The main limitations of the reported research should be briefly mentioned in the Discussion subsection.
